# Mesenchymal Stem-Cell Remodeling of Adsorbed Type-I Collagen—The Effect of Collagen Oxidation

**DOI:** 10.3390/ijms23063058

**Published:** 2022-03-11

**Authors:** Regina Komsa-Penkova, Galya Stavreva, Kalina Belemezova, Stanimir Kyurkchiev, Svetla Todinova, George Altankov

**Affiliations:** 1Department of Biochemistry, Medical University-Pleven, 5800 Pleven, Bulgaria; regina.komsa-penkova@mu-pleven.bg; 2Department of Experimental and Clinical Pharmacology, Medical University-Pleven, 5800 Pleven, Bulgaria; drstavreva@yahoo.com; 3Tissue Bank BulGen, 1330 Sofia, Bulgaria; kalina.belemezova@gmail.com (K.B.); kyurkch@hotmail.com (S.K.); 4Institute of Biophysics and Biomedical Engineering, Bulgarian Academy of Sciences, 1113 Sofia, Bulgaria; todinova@abv.bg; 5Research Group “Telemedicine and 3D Medicine”, Research Institute, Medical University-Pleven, 5800 Pleven, Bulgaria; 6Associate Member Institute for Biophysics and Biomedical Engineering, Bulgarian Academy of Sciences, 1113 Sofia, Bulgaria

**Keywords:** adipose tissue-derived mesenchymal stem cell, collagen type I, remodeling, oxidation

## Abstract

This study describes the effect of collagen type I (Col I) oxidation on its physiological remodeling by adipose tissue-derived mesenchymal stem cells (ADMSCs), both mechanical and proteolytic, as an in vitro model for the acute oxidative stress that may occur in vivo upon distinct environmental changes. Morphologically, remodeling was interpreted as the mechanical rearrangement of adsorbed FITC-labelled Col I into a fibril-like pattern. This process was strongly abrogated in cells cultured on oxidized Col I albeit without visible changes in cell morphology. Proteolytic activity was quantified utilizing fluorescence de-quenching (FRET effect). The presence of ADMSCs caused a significant increase in native FITC-Col I fluorescence, which was almost absent in the oxidized samples. Parallel studies in a cell-free system confirmed the enzymatic de-quenching of native FITC-Col I by Clostridial collagenase with statistically significant inhibition occurring in the oxidized samples. Structural changes to the oxidized Col I were further studied by differential scanning calorimetry. In the oxidized samples, an additional endotherm with sustained enthalpy (∆H) was observed at 33.6 °C along with Col I’s typical one at 40.5 °C. Collectively, these data support that the remodeling of Col I by ADMSCs is altered upon oxidation due to intrinsic changes to the protein’s structure, which represents a novel mechanism for the control of stem cell behavior.

## 1. Introduction

In addition to their regenerative activity, mesenchymal stem cells (MSCs) are highly involved in extracellular matrix (ECM) remodeling [1,2]. MSCs are multipotent stem cells (often referred to as adult stem cells) residing in most tissues poised to repair damage associated with trauma and ageing [3]. As such, they are of great interest for most cell-based therapies [4]. Adipose tissue-derived MSCs’ (ADMSCs) diverse differentiation capacity and immunomodulatory activity combined with their relative abundance, accessibility and low donor site morbidity render them particularly attractive therapeutic agents [5].

ECM remodeling describes the tightly controlled balance maintained between matrix protein formation and degradation. This highly dynamic process is central to tissue reconstruction during development, cell differentiation and various aspects of tissue homeostasis [6,7,8]. Abnormal ECM remodeling is characteristic of over 200 genetic and autoimmune conditions [9,10] in addition to distinct connective tissue disorders such as fibrosis and cancer [9]. Collagen remodeling is a cell-driven process that is critical during development, wound healing and regeneration and also involved in various pathological conditions, such as inflammation, scar formation, ageing and tumor progression [11].

Fibrillar collagen type I (Col I) is the most abundant type of collagen with over 29 types identified to date, classified into several groups according to the structures they form [12,13]. Col I comprises roughly 80–90% of the total collagen mass in the human body and provides most tissues and organs with shape, firmness, maturity, integrity and connectedness [12]. This naturally occurring biomaterial is used extensively as a substrate in tissue engineering and regenerative medicine due to its excellent biocompatibility, negligible immunogenicity, high biodegradability and favorable interactions with growth factors and cell adhesion molecules [12]. In addition, Col I is a ligand for specific cell receptors such as integrins, discoidin domain receptors, glycoprotein VI and the mannose receptor family, thereby controlling various important cellular activities including extracellular matrix (ECM) formation and turnover [14]. As a major protein component of the ECM, collagen contains a triple-helical domain with a unique periodical (Gly-X-Y)n structure in which X and Y are proline and 4-hydroxyproline, respectively [12].

The biosynthesis of collagen is a multistep process involving a number of post-translational modifications (PTMs) including chain association, folding, secretion, self-assembly and progressive cross-linking [15]. The PTMs depend on the oxidation of lysine and proline residues, as these are critical factors for the structural and biomechanical functions of Col I fibrils. The oxidation of lysine and proline occurs in response to distinct environmental changes [15], for example, as a part of physiological collagen processing during fibrillogenesis and osteocalcification. However, this might also be a component of oxidative stress [16,17], dependent on the production of reactive oxygen species (ROS) including free radicals and peroxides [16]. The ROS-induced oxidation of collagen and other ECM proteins modulates the ECM by altering its production, turnover and PTMs and, thus, strongly impacts cell–ECM interactions [18]. Oxidative stress is an important mediator in numerous pathological conditions including neurodegenerative diseases [19], cardiovascular complications [20], atherosclerosis [17,21] and ageing [22]. It also enables tissue repair after injury [23].

Despite the extensive investigations on the role of oxidative stress in collagen genes’ expression and collagen turnover related to various diseases [19,20,21], studies that directly utilize cellular models are relatively rare [24,25,26]. Though native collagen undergoes intensive remodeling by cells, particularly fibroblasts, the specific role of MSCs has been rather poorly investigated [12]. Despite comprehensive research on Col I processing by cells in 3D gel environments [27,28], substantially less effort has been dedicated to investigations with planar (2D) collagen-coated substrata. Nevertheless, in a few studies, the cell-dependent remodeling of adsorbed collagen is presented as three specific morphological events: (1) mechanical reorganization; (2) extracellular fibrils’ deposition; and (3) pericellular proteolytic degradation [2,29,30]. There are no investigations, however, on the behavior of stem cells in these conditions.

This study aimed to compare the biological response of ADMSCs adhered to native and pre-oxidized Col I substrata in an in vitro model of acute oxidative stress to determine how this common clinical occurrence affects stem cells’ behavior. Toward that end, we visualized adsorbed Col I in the adherent ADMSCs and developed a system to quantify the stem cell-driven proteolytic remodeling of collagen using fluorescent probes.

## 2. Results

The remodeling of adsorbed FITC-Col I was investigated using both morphological and quantitative approaches. Native Col I was labelled with FITC (FITC-Col I) and oxidized according to a previously described protocol (FITC-Col I OXI) [31]. For the morphological studies, ADMSCs were cultured for 24 h on glass coverslips pre-coated with either native FITC-Col I or FITC-Col I OXI. Thereafter, the cells were fluorescently stained with rhodamine conjugated phalloidin and Hoechest 33342 to illuminate, in different colors, the substratum-bound FITC-collagen (green), the actin cytoskeleton (red) and intact nuclei (blue) simultaneously. The second, quantitative approach involved measuring the de-quenching of the FITC-Col I fluorescent signal caused by cellular proteolytic activity, which leads to a proportional rise in the fluorescence (FRED effect) [32]. Collagenase CH (from *Clostridium histolyticum*) added to a cell-free system served as a positive control to confirm the de-quenching effect and compare the native and oxidized Col I’s susceptibility to proteolysis.

### 2.1. Overall Design of the Experiments

A diagram of the experiments is presented in Figure 1. The substrates were coated with FITC-Col I or FITC-Col I-OXI under standard protocol (100 μg/mL in 0.05 M acetic acid, incubated for 60 min at 37 °C) as detailed in the Materials and Methods section. Next, the cells were plated and allowed to adhere for 2 h in a serum-free medium to ensure the ADMSCs attached to collagen only. Afterwards, the medium was exchanged with one containing 10% serum and the cells were cultured for up to 24 h.

The subsequent steps for the morphological and quantitative evaluations of ADMSCs’ proteolytic activity are detailed below.

### 2.2. Morphology of ADMSCs Adhered to Col I or Col I-OXI

Figure 2 shows the data from a preliminary experiment to determine how cells recognize nonlabelled native and oxidized collagen. Phase-contrast images acquired after 2 h of incubation (A, B) depict slightly delayed ADMSC adhesion on oxidized samples, with more rounded but already well-attached cells, pointed to with a white arrow. However, this difference disappeared after 24 h of incubation, resulting in similar cell morphology and equally well-developed vinculin-positive focal adhesion complexes in both the native and oxidized samples (C, D, yellow arrows).

Figure 3 presents typical images of native (left panel) and oxidized (right panel) FITC-Col I substrata to which ADMSCs have adhered for 24 h. According to their actin cytoskeletons, the cells appeared to spread equally well on both substrata with similar polarized morphologies, spreading and actin cytoskeleton development (compare A and E). For clarity, the lower panels (B, F) contain images of the underlying substrates, with artificially superimposed cell contours and nuclei, while on (C, G), cell shapes are omitted. As shown on the left panel (A–C), when adhered to regular FITC-Col I, the ADMSCs tended to mechanically rearrange the underlying fluorescent layer into typical fibrillary assemblies located mostly beneath the cells (yellow arrows) and sparsely at the cells’ periphery (white arrows). On the oxidized samples (right panel), the fibrillary structures were missing (E–G). Relatively homogenous accumulation of the labelled protein beneath the cells was observed on both native and oxidized samples. D and H represent the typical views of the plane substrata, native and oxidized, respectively, with no cells added. No significant difference in the morphology of the adsorbed proteins was found at this magnification. Corresponding AFM images (see Appendix A) also showed a rather minor difference in the overall architecture of adsorbed protein layers, albeit the images on the augmented insets suggest better-expressed relief of native Col I at an ultrastructural level.

The ability of ADMSCs to form fibrillar collagen arrangements was quantified by counting the cells associated with fibers versus all cells in the sample. Quantifying 16 samples (6 native FITC-Col I and 10 with FITC-Col I OXI) revealed that 85.0 ± 18.03% of the cells adhering to native FITC-Col I were associated with fibers compared with only 11.22 ± 11.25% in the oxidized samples (*p* < 0.05) (Appendix A). This difference was confirmed by automated image analysis using the FibrilTool plug-in of ImageJ. Although surprisingly low anisotropy of the samples was observed, 0.037 ± 0.016 for the native FITC-collagen versus 0.007 ± 0.006 for the oxidized form, this difference was statistically significant (*p* < 0.05) (details are presented in a Appendix A).

### 2.3. Quantitative Measurement of FITC-Col I Binding

Col I was labelled with FITC according to Doyle [33] with some modifications, including adjusting the pH of the samples to pH 8 to ensure an effective FITC binding. The efficiency of FITC-collagen binding was calculated to be 9.365 ± 2.229 for the native FITC-Col I and 8.473 ± 2.898 for FITC-Col I OXI, respectively (Appendix A), approximately 10% less for the oxidized samples (*p* > 0.05).

### 2.4. Quantitative Measurement of Col I Absorption

In a separate experiment, we measured the efficiency of FITC-Col I’s adsorption to the bottom of wells (quadruplicated experiment) and found an insignificant (*p* > 0.05) drop in the adsorbed fluorescent signal from 70322.50 ± 4841.97 RPU for the native FITC-collagen to 62280.25 ± 4398.65 RPU for the oxidized samples, a difference of approximately 12% (Appendix A).

### 2.5. Quantitative Measurement of Col I Degradation

The preliminary experiment with a collagenase CH solution of FITC-Col I substrates resulted in a proportional rise of the fluorescent signal (FRET effect), as seen in Figure 4. This was because a part of the fluorophore was day-quenched (reducing the fluorescence, FRET effect) [32] and de-quenched, thus increasing the fluorescence upon proteolytic degradation of the collagen. This significant rise (*p* < 0.05) in the fluorescence upon collagenase digestions provided the basis for the quantitative studies on cell-derived collagen proteolysis.

### 2.6. De-Quenching of FITC-Col I by ADMSCs—Effect of Oxidation

Cell-derived proteolysis was investigated experimentally in quadruplicate samples of native FITC-Col I and oxidized FITC-Col I substrate coatings on 24-well glass-bottom TC plates cultured for 24 h in the presence or absence of ADMSCs. The rough data for the native FITC-Col I samples (−cells) showed that distinct amounts of protein desorbed spontaneously from the substratum, giving rise to a signal of 41.858 ± 3.368 RPU. In the presence of cells (+cells), the fluorescence increased to 44.496 ±13.685 RPU (~7% increase), confirming the de-quenching effect of ADMSCs. The same trend was observed for the FITC-Col I substratum: the signal intensity increased from 11.682 ± 508 RPU for the controls (−cells) to 13.253 ± 5.312 RPU for the samples (+cells), again confirming the proteolytic de-quenching of ADMSCs (by approximately 11%) but with values in the range of the signal scatter obtained for the supernatants. The oxidized Col I samples exhibited a similar pattern. Therefore, for ease of comparison, all data regarding ADMSC-derived proteolysis are presented as ΔRPU comparing the signal from the samples with cells (+cells) minus signal from samples (−cells).

Figure 5A presents the ΔRPU of the substratum-associated FITC-Col I (measured from the bottom of the wells) and Panel B contains those released in the medium. The left columns of both panels show the data for native FITC-Col I and the right columns reflect the oxidized samples. A significant increase in ΔRPU was observed for the substratum-associated native FITC-Col I (*p* < 0.05) versus almost no de-quenching for the oxidized samples. A similar, significant de-quenching (*p* < 0.05) was observed for the native FITC-Col I substrate released in the medium (Figure 5B) compared with the relatively small and nonsignificant increase (*p* > 0.05) in the oxidized samples.

The dramatic effect of oxidation is evident in the ΔRPU for native FITC-Col I (left columns, blue) and oxidized FITC-Col I (right columns, green). The fluorescence signal is presented as ΔRPU, representing the difference between samples (+cells) versus (−cells).

The observed difference in ADMSC-derived proteolytic remodeling could have been caused by two alternative mechanisms. First, the oxidized collagen may have triggered the differential secretion of matrix metalloproteases (e.g., thorough integrin-dependent inhibition), and second, the oxidation itself may have reduced the susceptibility of collagen to proteases. Therefore, a parallel experiment was designed to compare the degradation profiles of adsorbed native and oxidized FITC-collagen in a cell-free system (externally added collagenase without cells).

### 2.7. De-Quenching of FITC-Col I by Added CH Collagenase in a Cell-Free System

Figure 6 presents the ΔRPU for the native (blue line) and oxidized samples (green line) at 60 and 120 min of incubation in collagenase CH solution. While the native FITC-Col I was significantly de-quenched upon proteolytic digestion, the de-quenching was significantly less pronounced for the oxidized FITC-Col I probes.

We applied the Friedman test (Table 1) to confirm the significant increase in the fluorescence within the native FITC-Col I samples but not in the oxidized ones, proving that FITC-Col I-OXI is less susceptible to collagenase digestion.

These data suggest that the resistance to proteolysis of oxidized collagen samples observed in the cellular studies was caused by intrinsic changes in the collagen structure upon oxidation and not by altered cell signaling.

### 2.8. DSC Analysis of FITC-Col I–Effect of Oxidation

To evaluate the level of these structural changes, we investigated the effects of oxidation on the thermal stability of FITC-Col I by DSC analysis. DSC measures the heat capacity of samples as a function of temperature and provides information about the putative structural changes in the molecule.

DSC curves were also used to calculate thermodynamic parameters including the melting temperature (T_M_), the total transition enthalpy (∆H_total_) and the half-widths of transition (ΔT_M_ ½) (Table 2).

As expected, the maximum heat absorption of native FITC-Col I was observed at 40.5 °C (T_M-main_) (Figure 7, Table 2). In response to oxidation, the thermogram split into two well-resolved transitions with melting temperatures at 33.6 °C (T_M-pre_) and 40.1 °C (T_M-main_) (Table 2), which confirmed our previous investigation on calf skin collagen Type I that generated similar changes in collagen structure upon oxidation [31].

As shown in Table 2, the total enthalpy (∆H total) after oxidation was very close to that of native collagen, indicative of a discrete structural change in the collagen molecule upon oxidation, which was further confirmed by a slightly higher half-width of transitions in the oxidized sample compared with the native one.

## 3. Discussion

In most forms of their activity, cells tend to remodel their adjacent microenvironment via mechanical reorganization of ECM proteins and proteolytic degradation [34,35]. Both the morphological and quantitative studies presented here demonstrated that Col I underwent significant remodeling by stem cells; likewise, other matrix proteins (e.g., fibronectin, fibrinogen, vitronectin and type-IV collagen) were subjected to cellular remodeling [14,15,35,36,37,38,39,40]. These findings robustly reflect cells’ constitutive capacity to arrange their own ECM at the foreign material’s interface [35,36].

MSCs represent an important tool for tissue engineering, drug screening and disease modeling [3,4,41,42,43,44]. However, obtaining functional stem cells with controlled functionality still requires improvements as in vitro stem cells often lose their self-renewal and multi-lineage differentiation potential. The current strategies for maintaining MSC stemness (self-renewal and differentiation) largely focus on distinct ligand–receptor combinations, cell–cell adhesion (through N-cadherin) and soluble growth factors to stimulate differentiation [45,46,47]. However, the implementation of additional measures would greatly increase the likelihood of success [48,49]. In this context, the use of native collagen matrices is a challenging approach as collagen is a natural ECM protein with straightforward, tunable properties [40,44].

Relatively little is known on how stem cells behave in an acute oxidative environment [41,44]. Our data show unequivocally that oxidative conditions alter ADMSCs’ Col I remodeling; therefore, we anticipate that one such measure could be the modulation of cells’ oxidative environment.

ROS play a dual role in cell homeostasis [19,42]. Indeed, ROS are involved not only in the generation of oxidative stress and pathological environments but also mediate numerous cellular functions as secondary messengers [50,51,52]. The potential of ROS to regulate physiological processes was shown in a chondrogenic cell line (ATDC5), which experienced an increase in ROS over time in culture, thereby confirming its role in the chondrogenic response [41,47].

Recent data suggest that the ability to control the cellular redox status of their local microenvironment might be critical for MSC survival, expansion and differentiation [41,42].

Cell–substratum interaction is a complex process that is bi-directional and dynamic, mimicking the physiological interaction of cells with the ECM. Consequently, adherent cells tend to rearrange adsorbed ECM components [15,35,36,37,38]. Since cells likely employ a battery of adhesive proteins (e.g., fibronectin, vitronectin, fibrinogen, etc.) [35,36,37,38], to ensure that the cells attach exactly to the collagen in our system, we used collagen pre-coating followed by 2 h cell adhesion in a serum-free medium (Figure 1) before the serum was added (step B). Thus, we avoided the initial competitive effect of other serum proteins. In addition, fluorescently labelled collagen made it easier to visualize the cellular morphology and also to quantify the cellular proteolytic activity. Related to our experimental conditions, ADMSCs visibly recognize the native and oxidized Col I equally well at 24 h of incubation (when remodeling was studied), as evidenced by the lack of any significant difference in the overall cell morphology and focal adhesion formation (Figure 2 and Figure 3). The small delay in the initial cell attachment to oxidized samples observed after 2 h (Figure 2B) warrants further investigation that is outside of the scope of this study.

The ECM undergoes continuous proteolytic remodeling, which is a mechanism for the removal of excess ECM, a process often approximated with remodeling [15]. It must be noted, however, that cell-dependent ECM remodeling likewise includes the process of ECM organization and fibril formation, which is critical for their function and to interact with other cells [35,36]. Oxidative stress in vivo is characterized by a skewed ratio between collagen synthesis and degradation in which the former increases and the latter decreases [52]. The direct data on the degradability of oxidized collagen, however, are rather controversial. Chronic exposure to ROS leads to the fragmentation and accumulation of damaged collagen, rendering it more susceptible to proteolytic enzymes [52]. Cross-linking, however, inhibits collagen’s degradation [53], which stimulates MMP production [53,54]. Therefore, when considered in a broader sense, the capacity of cells to repair or replace ECM proteins following acute oxidative injury is likely to be a faithful predictor of how well cells respond to oxidative stressors [52].

The quantitative aspects of cellular proteolysis in vitro are also still debated. One approach to quantifying this activity, which was employed here, was to measure the increase in fluorescent signal that results from the proteolytic de-quenching of an initially day-quenched protein (FRET effect) [32,55]. Jadezko et al. established a protocol for ECM protein imaging in the presence of cells [32] adapted for confocal microscopy. Studies using the FRET effect to quantify cellular proteolysis in planar (2D) samples, however, are sparse and can largely be attributed to a line of our previous investigations [15,36,37,38].

An important observation concerning the proteolytic de-quenching of FITC-Col I is that it works well upon collagenase CH treatment, giving a significant rise in the total fluorescence (in a cell-free system). However, this was substantially inhibited in the oxidized FITC-Col I samples, indicating that oxidized collagen is more resistant to enzymatic degradation. We hypothesize that the effect of bacterial collagenase CH is related to the specific binding at its active site and its strong preference for glycine in P3 and P1′, proline at P2, and P2′ (according to the proteases’ classification) at its cleavage site [56]. The proline is particularly vulnerable to oxidation by metal ion-generated ROS and can be disproportionally modified by oxidation [57,58,59,60]; thus, the specific cleavage site could be lost.

We must consider, however, that morphological studies show that the space under the cells is not significantly affected and even some accumulation of FITC-Col I beneath the cells was observed in this study (Figure 3). Moreover, why FITC-Col I accumulate there, in both native and oxidized samples, remains unclear. One possible explanation is that the higher amount of FITC-Col I in the medium activates its transcytosis (vesicular transport of macromolecules from one side of a cell to the other) across the adherent cells [61]. Yet, we could not find any data on the transcytosis of collagen by MSCs, despite the link between protein transcytosis and ROS being demonstrated in other cell systems [62].

Nevertheless, our findings raise the question: why is oxidized collagen less sensitive to remodeling, both mechanical and enzymatic? We hypothesize that the process of collagenolysis depends on multiple interactions between mammalian collagenases and different exosites that serve to align the active site of collagenase, which possesses a strong preference for cleavage sites that perfectly match the repetitive amino acid sequence of the native Col I molecule [63]. These sites are located near the peptide bond to be cleaved, along with the local unwinding of the triple helix [63,64]. Any changes in the structure of collagen that result from oxidation could prevent it from being properly aligned with collagenase and, hence, spare its degradation. Moreover, the proline and hydroxyproline abundance at certain positions impacts the conformation of the collagen molecule, affecting even its affinity for integrin receptors [65]. However, the lack of a difference in the overall cell morphology after 24 h of incubation (after the slightly delayed initial attachment of ADMSCs to oxidized substrata) suggests that altered integrin activity is unlikely. Moreover, we found a strongly diminished susceptibility of oxidized collagen to collagenase in the cell-free system, which supports the view that intrinsic structural changes to the collagen molecule induced upon oxidation, but not altered cell signaling, are likely responsible for the effect.

To identify any putative structural changes to the Col I molecule, we performed DSC analysis comparing the native and oxidized samples (Figure 7), a follow-up of our previous investigation on calf skin collagen [31,66]. Here, we confirmed that the native FITC-Col I underwent a similar thermal transition change in response to heating, a single cooperative peak at 40.5 °C. The thermal denaturation of oxidized collagen, however, caused the main transition to split into two well-resolved transitions. Along with the above mentioned typical collagen endotherm, a new transition at 33.6°C appeared. Interestingly, the enthalpy (∆H) of the transition, which reflects its energetic aspects, was similar to that of native FITC-Col I. Since ∆H is dependent on the fraction of native protein in the solution [67], if this fraction is less in the total protein, ∆H would drop correspondingly [68], which was not the case here. It is also noteworthy that the transition half-widths (Table 2) were also close to those of the native sample, providing further indication that the observed pre-transition was caused by a rather discrete kind of damage of the collagen molecule. This allowed us to speculate that upon acute oxidation, the collagen molecule undergoes mostly intrinsic reorganization and does not go into the denatured state [66,67]. This again points to the possibility that the poor digestibility of collagen in an oxidizing environment depends on distinct changes to collagen’s structure that do not result from its denaturation.

Collectively, both the morphological and quantitative approaches demonstrated that native Col I underwent significant remodeling by stem cells. A completely novel observation, however, is that the oxidized collagen in acute conditions can be remodeled by ADMSCs neither mechanically nor enzymatically.

## 4. Materials and Methods

### 4.1. Collagen Preparation

Collagen type I (Col I) was isolated from rat tails using a standard procedure that combines acetic acid extraction and salting out with NaCl, as described previously [69,70]. Briefly, rat tails were obtained from the control animals of other experiments performed at Medical University Pleven, and tendons were gently removed from the tails, cleaned until free of tail debris and rinsed with distilled water and PBS. The fatty waste was removed by submerging the tails for 5 min in acetone and 70% isopropanol sequentially. Col I was solubilized in 0.5 M acetic acid with a magnetic stirrer at 4 °C for at least 48 h. After centrifugation at 4000 rpm at 4 °C, the supernatant was dialyzed extensively n 0.05 M acetic acid before the collagen was salted out with 1/5 its volume of 4.5 M NaCl. After centrifugation at 8000 rpm at 4 °C, the pellets were resuspended in 0.05 M acetic acid and dialyzed against 0.05 M acetic acid overnight. All procedures were run at 4 °C. Such collagen preparations are strongly enriched with type-I collagen [69,70]. The collagen concentration in the solutions was measured by modified Lowry assay [71] and by optical absorbance at 230 nm [72].

### 4.2. Fluorescent Labelling of Collagen

FITC-labelled collagen was prepared according to the modified protocol of Doyle [33]. Briefly, 4 mL of Col I solution in 0.05 M acetic acid (2.5 mg/mL) was titrated with 0.1M M borate buffer—pH (9.0) and mixed with 50 μL FITC dissolved in DMSO at 1 mg/mL, then incubated at room temperature for 90 min in the dark. The reaction was stopped with 0.05 M Tris buffer (pH 7.4) and any excess FITC was removed by intensive dialysis using 0.05 M acetic acid. Aliquots of FITC-labelled Col I were stored at +4 °C for up to 3 months

The molar ratio FITC/Protein (F/P) was calculated for collagen from UV-VIS spectral data of FITC-COL I and FITC-COL I-OXI using the adapted formula [73]:(1)$$F/P=F/C=\frac{\left(Amax\;\times \;D\right)}{\varepsilon ’\;\times \;{\;C}_{M}}$$
where A*max* is the absorbance of the FITC-COL I and FITC-COL I-OXI solutions, respectively, measured at 494 nm, the wavelength maximum (λmax) of FITC; D is a dilution factor; ε′ is the molar extinction coefficient of FITC, equal to 70,000 M^–1^cm^–1^; C_M_ is a molar collagen concentration.

### 4.3. Collagen Oxidation

FITC-Col I oxidation was performed according to a previously described protocol involving incubating the collagen solution (2 mg/mL) in 0.05 M acetic acid, at pH 4.3, with freshly prepared 50 µM FeCl_2_ and 5 mM H_2_O_2_ for 18 h at room temperature [31]. The reaction was stopped with EDTA at a final concentration of 10 mM. Any excess oxidants were removed by intensive dialysis against 0.05 M acetic acid.

#### Calculation of the Adsorption Rate of FITC-Col I and FITC-Col I-OXI

To calculate the adsorption rate of the collagen samples, 24-well glass-bottommed TC plates (Sensoplate, Greiner Bio-one, Meckenheim, Germany)) were pre-coated with-100 µg/mL native or oxidized FITC-Col I in 0.05 M acetic acid, incubated for 60 min at 37 °C, then washed 3 times with PBS before PBS was added (1 mL) to each sample for the fluorescent measurement. The measurement was performed at the excitation/emission wavelengths of 485/525 nm on the Multimode Microplate Reader (Mithras^2^ LB 943, Berthold Technologies GmbH & Co. KG, Bad Wildbad, Germany). All samples were analyzed in quadruplicate.

### 4.4. Cells

Human ADMSCs (passage 1) were obtained from Tissue Bank BulGen from healthy volunteers undergoing liposuction who provided written informed consent. The cells were maintained in DMEM/F12 medium containing 1% GlutaMAX™, 1% antibiotic-antimycotic solution and 10% fetal bovine serum (FBS), all purchased from (Thermo Fisher Scientific, Branchburg, NJ, USA). The medium was replaced every 2nd day until the cells reached approximately 90% confluence, at which point they were passaged.

### 4.5. Morphological Studies

For the morphological studies, standard (22 × 22 mm) glass coverslips (ISOLAB Laborgerate GmbH) were coated with FITC-Col I, either native or oxidized, dissolved in 0.05 M acetic acid (100 µg/mL, 60 min, 37 °C). The samples were placed on 6-well TC plates (Nunc, Denmark) and washed 3 times with PBS before being seeded with 5 × 10^4^ ADMSCs at a final volume of 2 mL in a serum-free medium (to assure the attachment to Col I only). To control the initial cell attachment after 2 h of incubation, the samples were monitored under phase contrast using an inverted microscope (Leica DM 2900). At the end of the second hour, the medium was replaced with one containing 10% serum, which also served to remove any unattached cells (less than 1%). The samples were further cultivated for up to 24 h, fixed with 4% paraformaldehyde and permeabilized with 0.5% Triton X-100 before fluorescence staining. To visualize the actin cytoskeleton, red fluorescent Rhodamine-Phalloidin (Invitrogen) was used (diluted 1:100 in PBS), while cell nuclei were stained with Hoechst 33342 (Sigma-Aldrich) diluted 1:2000 from a 10 mg/mL stock solution. For vinculin staining, a primary monoclonal anti-vinculin antibody (Sigma) followed by an Alexa fluor 555-conjugated goat anti-mouse secondary antibody (Sigma) was used. Finally, the samples were mounted upside down on standard glass slides with Mowiol (Sigma-Aldrich) and viewed through the green (FITC collagen), red (actin cytoskeleton, or vinculin) and blue (nuclei) channels of an upright fluorescent microscope (Olympus BX53) with UPlan FLN objectives at 40×0.50 magnification. The different colors were merged with Adobe Photoshop image processing software. At least three representative images were acquired for each sample.

#### 4.5.1. Quantification of Cell-Associated Fibrils of FITC-Collagen

The ability of ADMSCs to form fibrillar arrangements of collagen was quantified by determining the percentage of cells associated with fiber versus all cells in a given sample. For that purpose, images from 6 samples with native FITC-Col I and 10 with FITC-Col I OXI (16 samples total) collected from 4 independent experiments were investigated. All cells on the given image were counted through the red channel of a microscope (viewing actin cytoskeleton) and corroborated with those overlapping the fibrillary structures of FITC-collagen viewed through the green channel.

#### 4.5.2. Quantitative Morphological Analysis of Raw Format Images via FibrilTool Plug-In for ImageJ 

Qualitative data were gathered via the FibrilTool Java-build image postprocessing plug-in for ImageJ [74]. The anisotropy of fiber arrays and their average orientation was measured based on raw-format images of cells in a separate experiment performed under the same conditions. Images of equal size (W: 1600 px/H: 1200 px) were examined. The regions of interest (ROI) for the images of native FITC-Col I (4 ROI examined) and FITC-Col I-OXI (6 ROI examined) were delineated, with a summarized ROI area of the samples with FITC-Col I (1.0052 × 10^6^) and FITC-Col I -OXI (7.4105 × 10^5^). The average orientation and anisotropy of the fibrillar structures in each ROI were measured. Details are presented in Appendix A.

### 4.6. FITC-Collagen Degradation in a Cell-Free System

Quadruplicated samples of FITC Col I and FITC Col OXI coated substrata were produced in 24-well glass-bottommed TC plates (Sensoplate, Greiner Bio-one, Meckenheim, Germany). A standard protocol (incubation for 1 h with 100 μg/mL FITC-collagen solution in 0.05 M acetic acid followed by 3 PBS washes) was used to coat the plates. Next, collagenase type I from *Clostridium histolyticum* (Genaxxon Bioscience GmbH. Ulm., Baden-Württemberg, Germany) at 3.7 mg/mL in TC medium was added to the samples, which were then incubated for 60 and 120 min at 37 °C. The fluorescence of the quadruplicated samples was measured with the Multimode Microplate Reader (as above) set to excitation/emission wavelengths of 485/525 nm. Fluorescence intensity is presented either directly as relative photometric units (RPUs) or as ΔRPU (reflecting the difference in the fluorescent signal between samples with collagenase and the controls without collagenase).

### 4.7. FITC-Collagen Degradation by ADMSCs

To measure the cell-dependent proteolytic activity, the same approach of FITC-Col I de-quenching was used. Twenty-four-well glass-bottom TC plates were pre-coated as described above with 100 µg/mL native or oxidized FITC-Col I and washed 3 times with PBS before ADMSCs (1 × 10^4^ per well) were plated in a final volume of 1 mL of serum-free medium. After 2 h of incubation in this serum-free medium (ensuring the single-protein adhesion of the cells to Col I), the medium was exchanged with one containing 10% serum and the cells were cultured for up to 24 h in a humidified CO_2_ incubator. Then, the supernatants were collected to measure the released FITC-Col, while the adsorbed (substratum-associated) FITC-Col I was measured directly from the bottom of the plate (in 1 mL PBS) using the microplate reader (as above) set at 485/525 nm. Matched samples without cells (−cells) were processed in the same way and all experiments were quadruplicated. The measured fluorescence intensity is presented directly in RPU or as ΔRPU (representing the difference between samples (+cells) versus (−cells)) for the quantification of the ADMSC-dependent proteolytic de-quenching of FITC-Col I.

### 4.8. DSC Measurements

DSC measurements were performed using DASM4′s (Privalov, BioPribor, Moscow, Russia) built-in, high-sensitivity calorimeter with a cell volume of 0.47 mL. The samples were diluted in 0.05 M acetic acid before the DSC. The protein concentration was adjusted to 2 mg/mL. To prevent any degassing of the solution under study, constant pressure of 2 atm was applied to the cells. The samples were heated at a scanning rate of 1.0 °C/min from 20 °C to 65 °C and were preceded by a baseline run with buffer-filled cells. Each collagen solution was reheated after cooling from the first scan to evaluate the reversibility of the thermally induced transitions. The calorimetric curve corresponding to the second (reheating) scan was used as an instrumental baseline and was subtracted from the first scans, as collagen thermal denaturation is irreversible. The obtained excess heat capacity profiles were normalized to the protein concentration. The calorimetric data were analyzed using the Origin Pro 2018 software package.

### 4.9. Statistical Analysis

Data were analyzed using SPSS Statistics for Windows, Version 23.0 (IBM Corp, Armonk, NY, USA). All quantitative results were obtained from at least four samples. Descriptive data were compared using the Chi-square and Mann–Whitney U tests. The non-parametric differences between groups were compared using Friedman’s test; pair-wise comparisons were achieved using the Dunn–Bonferroni post hoc analysis. Data were expressed as mean ± standard deviation (SD). Differences with *p* < 0.05 were considered to be statistically significant.

## Figures and Tables

**Figure 1 ijms-23-03058-f001:**
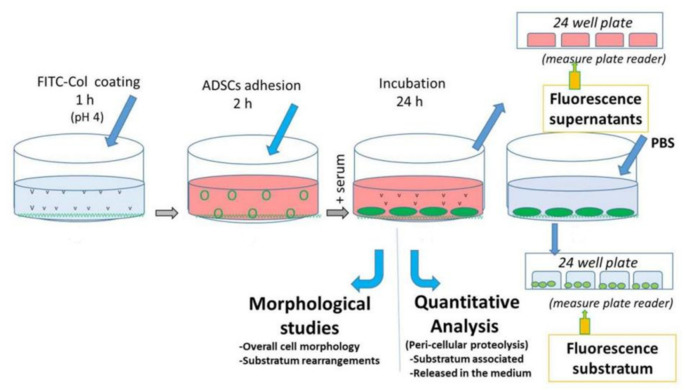
Schematic overview of FITC-Col I remodeling experiment.

**Figure 2 ijms-23-03058-f002:**
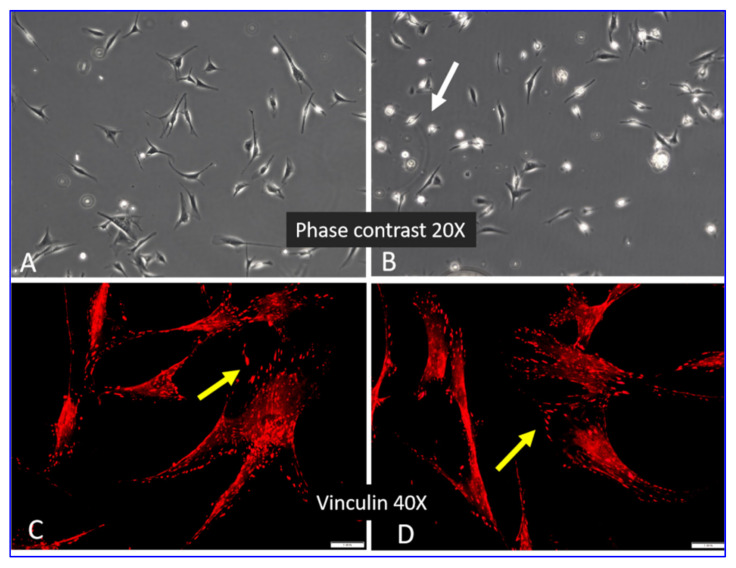
Initial adhesion at 2 h (**A**,**B**) and subsequent spreading after 24 h (**C**,**D**) of ADMSCs adhered to Col I (**A**,**C**) and Col I-OXI (**B**,**D**). Top row-cells imaged under phase contrast (20×); bottom row-focal adhesions viewed by staining for vinculin (40×). Scalebar 20 μm.

**Figure 3 ijms-23-03058-f003:**
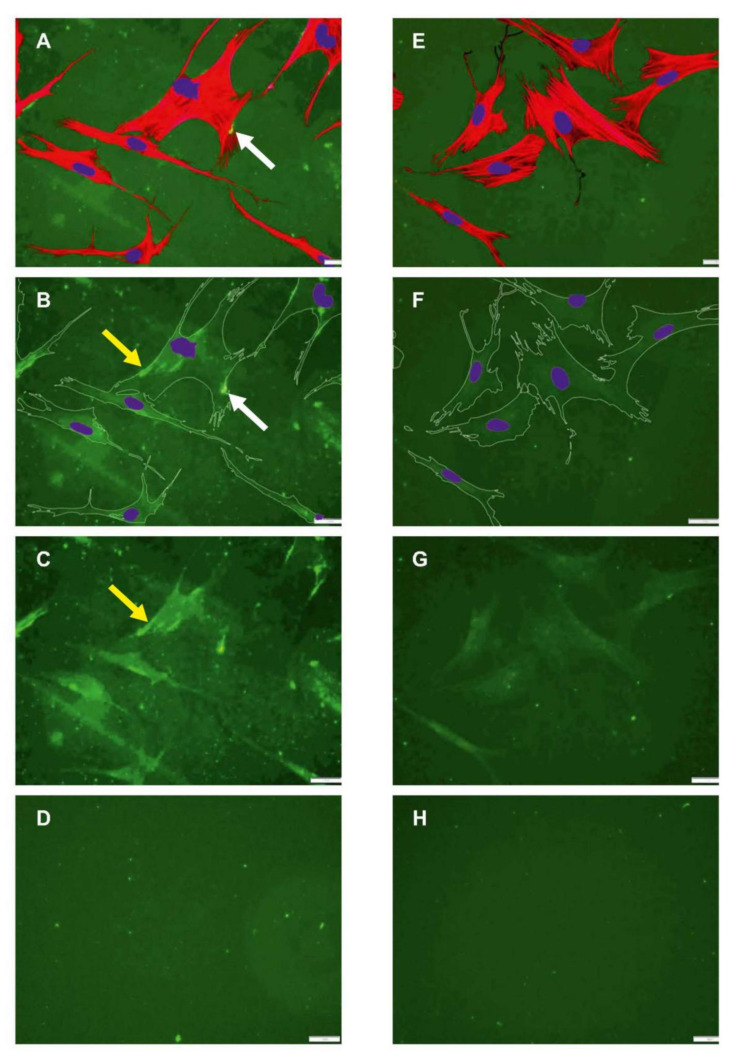
Morphology of ADMSCs adhering to native (**A**–**C**) and oxidized (**E**–**G**) FITC-Col I. The cells were stained for actin (red) and nuclei (blue), and viewed with the underlying fluorescent substratum (green). (**A**,**E**) present the overall morphology of cells adhering on native and oxidized collagen, respectively. (**B**,**F**) present the underlying substrates of the same samples with artificially superimposed cell contours and nucleus. In (**C**,**G**), the cell shapes are omitted. (**D**,**H**) show the native and oxidized collagen substrates, respectively, with no cells added. The yellow arrows point to typical fibrillary collagen arrangements beneath the cells, while the white arrows point to arrangements outside of the cells along the cells’ periphery. Scalebar 20 μm.

**Figure 4 ijms-23-03058-f004:**
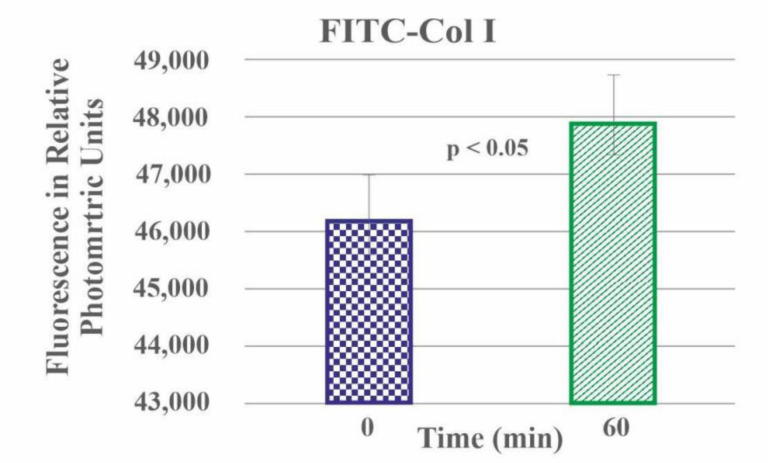
De-quenching of FITC-Col I in the presence of CH collagenase for 1 h at 37 °C.

**Figure 5 ijms-23-03058-f005:**
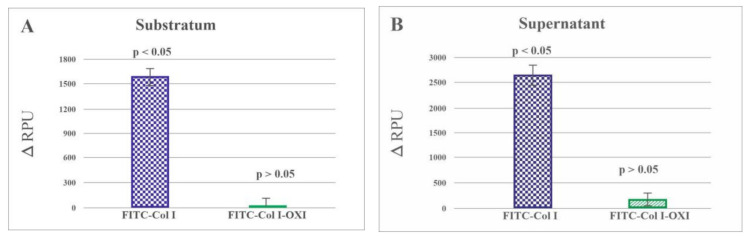
Relative changes in the fluorescence values of substratum-associated (**A**) and spontaneously released (**B**) FITC-Col I after 24 h of incubation with and without cells. The data are presented as RPU subtracting the signal from the samples with cells from the cell-free ones, directly characterizing the proteolytic de-quenching caused by the cells.

**Figure 6 ijms-23-03058-f006:**
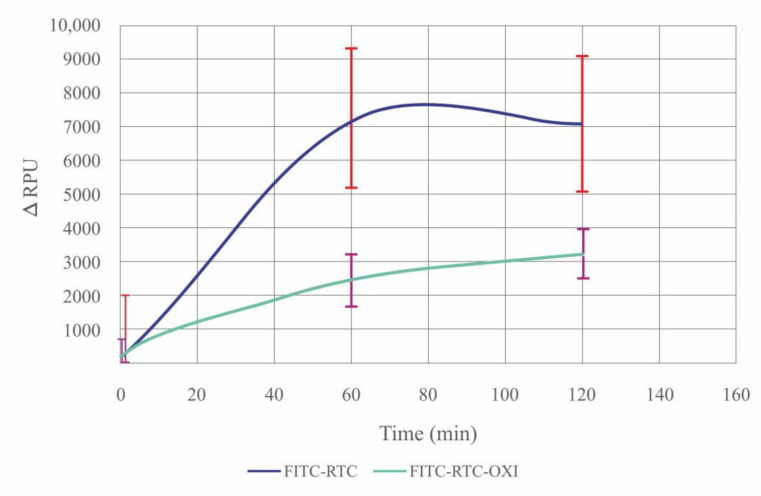
Relative changes in the fluorescence intensity upon addition of FITC-Col (blue) or oxidized FITC-Col I-OXI (green) to a solution of collagenase CH as substrates for proteolytic digestion at times of 0, 60 and 120 min at 37 °C. Fluorescence intensity is presented as RPU subtracting the fluorescent signal of samples with collagenase and those without collagenase (negative control), directly characterizing the proteolytic de-quenching.

**Figure 7 ijms-23-03058-f007:**
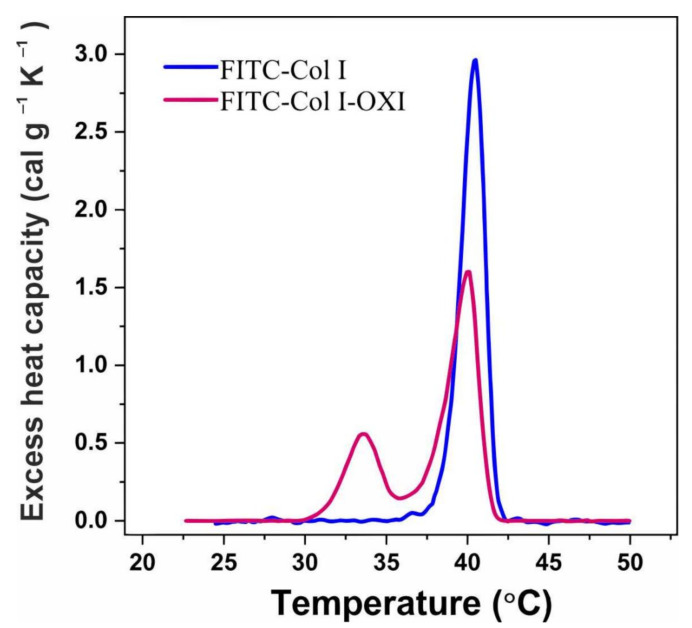
DSC thermograms of native FITC-Col I (blue line) and oxidized FITC-Col I-OXI (red line).

**Table 1 ijms-23-03058-t001:** Chi-Square and statistical significance of the relative changes in the fluorescence intensity (ΔRPU) upon collagenolytic action of collagenase CH on FITC-Col and oxidized FITC-Col I-OXI according to the Friedman test statistics.

Samples	FITC-Col I	FITC-Col I-OXI
Chi-Square	12.000	4.000
Sig. (p)	0.002	0.135

**Table 2 ijms-23-03058-t002:** Thermodynamic parameters—transition temperature (T_M_), total calorimetric enthalpy (∆H_total_) and transition half-widths (Δ T_M_ ½)—obtained from DSC profiles of FITC-Col I and FITC-Col I-OXI.

Collagen	T_M-pre_ (°C)	T_M-main_ (°C)	∆H_total_ (cal/g)	Δ T_M-main_½ (°C)
FITC-Col-I	-	40.5	5.58	1.72
FITC-Col I-OXI	33.6	40.1	5.50	2.09

T_M-main_—temperature of the main transition; T_M-pre_—temperature of the additional pre-transition event; ∆H_total_ —total transition enthalpy; Δ T_M-main_½—half-width of the main transition.

## Data Availability

Data available on request from G.Altankov and R. Komsa-Penkova.

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
