# Peer review of "Mesenchymal Stem-Cell Remodeling of Adsorbed Type-I Collagen—The Effect of Collagen Oxidation"

_ijms, 2022, doi:10.3390/ijms23063058_

Round 1

Reviewer 1 Report

This study whether oxidized collagen is I remodeled to the same extent as non-oxidized collagen I upon adhesion of adipose-derived mesenchymal stem cells. The main conclusion of the study is that oxidized collagen is less pliable than wild type collagen.

In reviewing a previous version of this manuscript, this reviewer identified fundamental issues that precluded publication then. I’m happy to see that the authors have addressed some of them, making this paper a stronger candidate. However, some issues remain.

  • Can oxidized collagen be made fluorescent to the same extent as native collagen? This is easy to test and needs to be tested.
  • Likewise, is oxidized collagen equally assembled in the absence of cells? The authors need to present evidence that the matrices are comparable in the absence of cells.
  • In nature, some fibers will be more oxidized than others. Authors could label native collagen with FITC, oxidized collagen with TRITC or similar, and combine them. According to their results, one would expect only native fibers to be remodeled, lending further support to their hypothesis. As is, the differences in adherence and structure per se can be too dramatic.
  • What’s the evidence that collagen-binding integrins differentially bind oxidized collagen? This must have been addressed.
  • The de-quenching experiments mean that ADMSC do not remodel oxidized collagen. How is this different from a zymogram? It is possible that the different types of collagen trigger different degradative conditions in ADMSC. This needs to be tested.

Author Response

Dear Ms. Ternence Chen,

Dear Mrs. Editor,

We appreciate the opportunity to resubmit our manuscript entitled “Mesenchymal stem cells remodeling of adsorbed type I collagen— the effect of collagen oxidation” to be considered for publication in Int. J. Mol. Sci.

We would also like to express our thanks to the reviewers for the very helpful comments and suggestions. Accepting most of them we believe it has resulted in an improved version of the manuscript. We enclose the Point-by-point response to reviewers’ comments for your consideration.

Point-by-Point response to the reviewer’s comments.

Reviewer 1

Comments and Suggestions for Authors

This study whether oxidized collagen I is remodeled to the same extent as non-oxidized collagen I upon adhesion of adipose-derived mesenchymal stem cells. The main conclusion of the study is that oxidized collagen is less pliable than wild-type collagen.

In reviewing a previous version of this manuscript, this reviewer identified fundamental issues that precluded publication then. I’m happy to see that the authors have addressed some of them, making this paper a stronger candidate. However, some issues remain.

Answer:

In our opinion, we addressed all the issues raised by the reviewers finding them relevant and very helpful to improve the manuscript.

  • Can oxidized collagen be made fluorescent to the same extent as native collagen? This is easy to test and needs to be tested.

Answer:

In this study, the oxidation of collagen was performed after FITC labeling (Materials &Methods, 4.2 and 4.3), clearly emphasized in our previous revision.  However, as suggested by the estimated Reviewer 1, we performed additional experiments (supplementary table 1 and 2) showing that oxidation of FITC collagen did not change significantly its fluorescence, and its substratum-binding properties. In respect to the post-oxidative labeling of collagen, i.e. whether the oxidation would affect the FITC binding itself, we have to answer that in the preliminary experiments aimed to check this alternative situation, we observed that the labeling of oxidized collagen significantly increases FITC binding (about 30%) and also reduces its adsorption to plates, presumably due to some structural changes in the collagen upon conjugation. Thus it become out of the scope of this study and we strictly settled on the protocol for FITC labeling before oxidation to avoid any “additive” FITC binding effects that may compromise the results, particularly the de-quenching by proteases. We did not include this result in the paper because it concerns conditions that are not relevant to the conditions used in all other experiments.

  • Likewise, is oxidized collagen equally assembled in the absence of cells? The authors need to present evidence that the matrices are comparable in the absence of cells.

Answer:

We acknowledge this important comment. The images provided in Fig 3 (D and H) of the plane substrate (without cells) show no significant difference in the assembly of FITC Col I and FITC Col I OXI upon adsorption on substratum, suggesting that the structural changes are rather discrete. Minor changes in the structure of the adsorbed oxidized collagen were observed also by AFM studies (performed recently in connection with another line of our research). These AFM images suggest better-structured relief of native Col I as compared to oxidized one, which is expectable, taking in mind the data from DSC analysis (Figure 7) and the difference in the biological response of stem cells (abrogated remodeling). To enrich this aspect of the study, we provide these AFM data as a supplementary Figure 1S to this paper. Accordingly, we add a respective comment in the Results section at the end of the first paragraph (page 6, lines 160-163) as follows:

“Corresponding AFM images (see supplementary Figure 1S) also showed rather a minor difference in the overall architecture of adsorbed protein layers; albeit the images on the augmented insets suggest better-expressed relief of native Col I at an ultrastructural level.”

The following text is added along with the new Figure 1S.

Figure 1S. Representative 2D AFM images of native FITC collagen (A) and oxidized FITC collagen (B); 3D topographical images of the images in A and B. The insets in A and B represent enlarged images of an area with the size of 0.8x0.8 µm of the same panel. The AFM imaging was carried out using Atomic Force Microscope Asylum Research MFP-3D (Oxford Instruments) operated in contact mode at room temperature. Silicon nitride probes (Nanosensors, type qp-Bio) with a spring constant of 0.06 N/m, resonant frequency 16 kHz, and a nominal tip radius of 8 nm were used in AFM measurements. Morphological characteristics were analyzed using IgorPro software, embedded in the AFM system. The AFM imaging was done in Institute of Optical Materials and Technologies, Bulgarian Academy of Sciences.”

  • In nature, some fibers will be more oxidized than others. Authors could label native collagen with FITC, oxidized collagen with TRITC, or similar, and combine them. According to their results, one would expect only native fibers to be remodeled, lending further support to their hypothesis. As is, the differences in adherence and structure per se can be too dramatic.

Answer:

We appreciate this interesting suggestion; unfortunately, a time restriction of 10 days provided by the Editor for this revision precludes designing a new experiment. Moreover, fluorescent labels have different binding efficiency, which raises additional concerns. What we can share here is that we tried to use another fluorescent marker Atto 488, but with limited success. So, we finally stop on FITC, because of our previous experience and because of the moderate costs of this labeling procedure. But we keep in mind the original experimental design suggested by the estimated reviewer.

  • What’s the evidence that collagen-binding integrins differentially bind oxidized collagen? This must have been addressed.

Answer:

At the 24th hour of incubation, we did not observe any difference neither in the overall cell morphology nor in the formation of focal adhesion contacts between oxidized and native collagen samples, making the hypothesis for different integrin-binding unlikely. It is evident from Figure 2 and Figure 3. Moreover, the observed phenomena of reduced proteolytic remodeling in oxidized samples work also in a cell-free system, with externally added bacterial collagenase, where no integrin activity is involved. We stress this issue in the Results section (paragraph 2.6, page 8, lines 227-233), and in the last (conclusive sentence) of paragraph 2.7 (page 8  lines 257-259).

  • The de-quenching experiments mean that ADMSC does not remodel oxidized collagen. How is this different from a zymogram? It is possible that the different types of collagen trigger different degradative conditions in ADMSC. This needs to be tested.

Answer:

Zymography is an electrophoretic method for measuring proteolytic activity based on SDS gel impregnated with a given protein substrate. We consider using this approach to identify which exactly collagenases are involved in the cellular effect. But we have to note that it is a rather qualitative approach (though the white bands might be semi-quantified, admitting that a certain range of band intensity relates linearly to the amount of protease loaded). However, this is again beyond the scope of this study, as here we describe a more general property of oxidized collagen to be less accessible for collagenases, evident both in the presence of ADMSC and in a cell-free system.

*Note: According to the corrections of the English language editor we changed the title of the manuscript  ….

Original one: “Mesenchymal stem cells remodeling of adsorbed type I collagen— effect of collagen oxidation”

Corrected: “Mesenchymal stem cells remodeling of adsorbed type I collagen— the effect of collagen oxidation”

Submission Date

05 March 2022

Reviewer 2 Report

The authors showed the results that oxidation of collagen affects the proteolysis and the ADMSCs morphology. However, the oxidation of collagen per se change the nature of collagen, which was shown by DSC thermograms.

Potentially this manuscript contains precious information.

Major comments

Figure 3, FITC-Col ADMSCs seem to have more green intensity than FITC-Col XI. Does this mean ADMSCs absorb more collagen? If so, why the authors did not intensify this result?

Figure 5 and 6 may be reversed.

The results are simple, but the writing is too complicated. 

If the schematic figure of the results is presented, it may be more understandable. 

Author Response

Dear Ms. Ternence Chen,

Dear Mrs. Editor,

We appreciate the opportunity to resubmit our manuscript entitled “Mesenchymal stem cells remodeling of adsorbed type I collagen— the effect of collagen oxidation” to be considered for publication in Int. J. Mol. Sci.

We would also like to express our thanks to the reviewers for the very helpful comments and suggestions. Accepting most of them we believe it has resulted in an improved version of the manuscript. We enclose the Point-by-point response to reviewers’ comments for your consideration.

A point-by-Point response to the reviewer’s comments.

Reviewer 2

Comments and Suggestions for Authors

  • The authors showed the results that oxidation of collagen affects the proteolysis and the ADMSCs morphology. However, the oxidation of collagen per se changes the nature of collagen, which was shown by DSC thermograms.

Answer:

Yes, oxidation changes the natural structure of collagen and it is particularly evident from DSC measurement. Presumably, these structural changes are not so dramatic (rather discrete), but they may alter the collagen processing in the body, at least under certain oxidative conditions.  Actually, this is the main message of this paper.

  • Potentially this manuscript contains precious information.

Answer:

Thank you for this positive assessment.

Major comments

  • Figure 3, FITC-Col ADMSCs seems to have more green intensity than FITC-Col OXI. Does this mean ADMSCs absorb more collagen? If so, why the authors did not intensify this result?

Answer:

This was also our first assumption, but it is rather accidental and differs from picture to picture. The quantitative studies provided in supplementary Table 1S did not confirm significantly such difference, i. e. less fluorescence of FITC-Col I OXI samples, though some tendency may be observed.

  • Figures 5 and 6 may be reversed.

Answer:

Thank you for the noted reverse. It was a technical error, now corrected. However now we observed that during a previous revision the legends to these figures become a bit heavy. Now we slightly shorten them as follows:

Figure 5. Relative changes in the fluorescence values of substratum-associated (A) and spontaneously released (B) FITC-Col I after 24 h of incubation with and without cells. The data are presented as DRPU subtracting the signal from the samples with cells from the cells-free ones, characterizing directly the proteolytic de-quenching caused by the cells.

Figure 6. Relative changes in the fluorescence intensity upon addition of FITC-Col (blue) or oxidized FITC-Col I-OXI (green) to a solution of collagenase CH as substrates for proteolytic digestion at times 0, 60 and 120 min at 37°C. Fluorescence intensity is presented as DRPU subtracting the fluorescent signal of samples with collagenase and those without collagenase (negative control), characterizing directly the proteolytic de-quenching.

  • The results are simple, but the writing is too complicated.  If the schematic figure of the results is presented, it may be more understandable.

Answer:

This is true, the experimental presentation becomes heavy. Therefore we made an explanatory scheme as Fig 1 at the beginning of the Results section. Probably it is not sufficient. We are afraid, however, that if we provide an additional scheme (with final results) it would make the section heavier.  Actually, we believe that the Graphic abstract (provided) should play a similar role.

*Note: According to the opinion of the English language editor we changed the title of the manuscript  ….

Original one: “Mesenchymal stem cells remodeling of adsorbed type I collagen— effect of collagen oxidation”

Corrected: “Mesenchymal stem cells remodeling of adsorbed type I collagen— the effect of collagen oxidation”

Submission Date

05 March 2022

Round 2

Reviewer 1 Report

I agree that the authors should have been given more time to address my comments. With the editorial timeline given to them, you have done your best and provide cogent responses to my questions. I have no further queries that can be addressed within its timeframe.

Reviewer 2 Report

I still think that a schematic summary of the results will help to understand the results more clearer. However, this is about the presentation, not the essential problem.

This manuscript is a resubmission of an earlier submission. The following is a list of the peer review reports and author responses from that submission.

Round 1

Reviewer 1 Report

This study addresses the effect of oxidation on collagen remodeling by adipose-derived mesenchymal stem cells. The main conclusion of the study is that oxidized collagen is way harder to remodel than wild type collagen.

This study contains fundamental flaws that preclude its publication. In addition, the copy is poorly edited, including numerous typos and authors’ notes that make for a cumbersome, sloppy read.

The following comments are offered to help the authors as they submit their research elsewhere.

  • The authors do not consider the effect of collagen oxidation on the efficiency of FITC binding. If oxCOL binds FITC as well as natCOL, it needs to be formally demonstrated.
  • The authors do not consider the effect of collagen oxidation on the efficiency of collagen binding to the culture surface as depicted in Figure 1. If oxCOL binds as well as natCOL to the surface, it needs to be measured and those data included.
  • Extracellular matrix proteins in 2D are notoriously different from that in 3D. Based on the experimental design of the authors, 3D experiments can be easily carried out, adding a lot of basic information.
  • 2 is completely unacceptable by today’s standards. A systematic (hundreds of cells from different experimental replicas) analysis is required, along with quantification of the observed phenomena.
  • If oxCOL not remodeled by ADMSC to the same extent as natCOL, the effect on cell adhesion should be evident, and easily revealed by staining for focal adhesion components.

The de-quenching experiments shown in Figs. 3-5 are interpreted as ADMSC do not remodel oxCOL as well as they do natCOL, but other interpretations are possible, including that integrin-based activation of ADMSC trigger the differential secretion of matrix metalloproteases, which is likely in the time frame of the experiments. The experiments should be done in conditions in which secretion is hampered, or taken into account at least. 

Reviewer 2 Report

The manuscript by Komsa-Penkova and colleagues aims to evaluate the effect of oxidation in MSC collagen remodeling. This work arises as a continuation of the group research in this field published a few years ago. 

I recognize the value of this work nevertheless I believe there are major issues preventing the publication of this work as is. I would recommend the authors to address the following points:

  • Figure 2 is as a preponderant result from this work. However the images presented are insufficient to convince this reader regarding the assumptions made. The authors obtained three representative images acquired per sample (line 469). The events highlighted by arrows are not significantly representative. I would recommend a repetition of this study and the randomly acquisition of several photos for morphometric analysis. I suggest the quantification of Col fibrillary assemblies in native vs oxidized samples.
  • Following the previously explained I consider the statements in lines 144-149, 243, 246-248 and 292-295 to be abusive interpretation of the results obtained and the data shown. 
  • Regarding the methodology of these trials, where 2 hours without serum sufficient to allow cell adhesion? How did you check this? Prior serum supplementation, did you perform any cell culture wash to remove eventual floating cells? If so, please explain and add to the methods section. 
  • Lines 413-415, the authors state a new mechanism for MSC control was found via subtle changes in oxidative environment. 1. Is the in vitro model used a mimic of “subtle” oxidative changes? 2. Please refrain the statement “encountered a novel mechanism”. 

Other issues:

  • Results section often includes unnecessary information repeated from materials and methods. 
  • Figure 3 does not show the results from 120 min as mentioned in lines 163-167.
  • Please clarify the point (B), line 173-174, in Figure 4
  • Discussion is rather long. Some information is not fully relevant regarding the data presented, for instance paragraph in lines 355-376.
  • Line 317 - note reference formatting